



# Comparison of large-scale dynamical variability in the extratropical stratosphere among the JRA-55 family data sets

Masakazu Taguchi[1]

[1]Department of Earth Science, Aichi University of Education, Kariya, 448-8542, Japan

*Correspondence to*: Masakazu Taguchi (mtaguchi@auecc.aichi-edu.ac.jp)

**Abstract.** This study compares large-scale dynamical variability in the extratropical stratosphere, such as major stratospheric sudden warmings (MSSWs), among the Japanese 55-year Reanalysis (JRA-55) family data sets. The JRA-55 family consists of three products: a standard product of the JRA-55 reanalysis data, and two sub-products of JRA-55C and JRA-55AMIP. JRA-55C assimilates only conventional observations, whereas JRA-55AMIP runs the same numerical weather prediction

model without assimilation of observational data. A comparison of the occurrence of MSSWs in Northern winter shows that compared to the standard product, JRA-55C delays several MSSWs by one to four days and also misses a few MSSWs. JRA-55C also misses the Southern Hemisphere MSSW in September 2002. JRA-55AMIP shows much fewer MSSWs in Northern winter, and especially lacks MSSWs of high aspect ratio of the polar vortex. A further examination of daily geopotential height differences between JRA-55 and JRA-55C reveals occasional peaks in both hemispheres. The delayed

and missed MSSW cases have smaller height differences in magnitude than such peaks. The differences include large contributions from the zonal component, which are consistent with underestimations in the weakening of the zonal mean polar night jet in JRA-55C. We also explore strong planetary wave forcings and associated polar vortex weakenings for JRA-55 and JRA-55AMIP. It shows a lower frequency of strong wave forcings and weaker vortex responses to such wave forcings in JRA-55AMIP, consistent with the lower MSSW frequency.

**1 Introduction**

Large-scale dynamical variability is a vital feature in the extratropical stratosphere especially in the Northern Hemisphere (NH) winter stratosphere (e.g., Labitzke and van Loon, 1999; Yoden et al., 2002; Waugh and Polvani, 2010). The NH winter stratosphere exhibits large intraseasonal and interannual variations, reflecting anomalously strong and weak conditions of the polar vortex. The polar vortex in the Southern Hemisphere (SH) spring stratosphere also shows large variations, e.g., in its

strength and distribution, while the SH winter stratosphere is more dynamically quiescent.

Some of weak conditions of the polar vortex correspond to the occurrence of stratospheric sudden warmings (SSWs), during which the polar night jet weakens and polar stratospheric temperatures rise as the polar vortex largely distorts and/or breaks down (e.g., Limpasuvan et al., 2004; Charlton and Polvani, 2007). A SSW is classified as a major SSW (MSSW), when it





accompanies a reversal of the zonal mean zonal wind often looked at $60^{\circ}$ N, 10 hPa (e.g., Butler et al., 2015). A SSW without such a zonal wind reversal is classified as a minor SSW.

Previous studies investigate aspects of such dynamical variability in the extratropical stratosphere using multiple reanalysis data sets. Reanalysis data sets are a vital tool to understand atmospheric variability and relevant processes in the climate

science including the middle atmosphere science, but different reanalyses sometimes yields different results for the same diagnostics. Martineau and Son (2010) examined time evolutions of the Northern Annular Mode index for stratospheric vortex weakening and intensification events among five reanalyses and found good agreement. Martineau et al. (2016) investigated dynamical consistency in the extratropical stratosphere among eight reanalyses as quantified by the residual of the zonal momentum equation. Applying a multiple linear regression analysis to nine reanalyses, Mitchell et al. (2015)

studied signatures of interannual variability associated natural forcings, and found remarkable similarity among the data sets. Manney et al. (2005) conducted a diagnostic comparison of the SH MSSW in September 2002 among several metrological data sets.

Some of these studies go along with a coordinated activity of the Stratosphere-troposphere Processes And their Role in Climate (SPARC) Reanalysis Intercomparison Project (S-RIP; Fujiwara et al., 2017). The climatology and interannual

variability of monthly mean temperature and wind fields are surveyed in the S-RIP framework. Furthermore, various aspects of the dynamical coupling between the NH extratropical stratosphere and troposphere will be also investigated.

This study focuses on the Japanese 55-year Reanalysis (JRA-55; Kobayashi et al., 2015) data among others, which is one of newer reanalyses. A unique feature of JRA-55 is that in addition to the standard product, two companion products, JRA-55C and JRA-55AMIP, are also available. JRA-55C assimilates conventional surface and upper air observations only, without

assimilation of satellite observations. JRA-55AMIP is an AMIP-type simulation using the same forecast model as in JRA-55 and JRA-55C, without assimilation of any observational data. The three products are called the "JRA-55 family data sets" as a whole (Kobayashi et al., 2014).

Previous studies investigated some aspects of the JRA-55 family data sets especially in the stratosphere. Kobayashi and Iwasaki (2016) examined the Brewer-Dobson circulation in the lower stratosphere to show that the mass stream function at

100 hPa (vertically integrated northward mass flux above 100 hPa) is similar between JRA-55 and JRA-55C in annual and seasonal averages, but it is much weaker for JRA-55AMIP. Kobayashi et al. (2014) and Kobayashi and Iwasaki (2016) also showed that the polar night jet in the winter stratosphere for each hemisphere is stronger for JRA-55C than for JRA-55, associated with weaker upward wave propagation and driving. Differences of JRA-55AMIP from JRA-55 are qualitatively similar, but are much larger in magnitude.

This study seeks to compare the climatology and large-scale dynamical variability, such as frequency and vortex geometry of SSWs, in the NH and SH extratropical stratosphere among the JRA-55 family data sets. A motivation for this study is that dynamical variability in the extratropical stratosphere remains relatively unexplored in the JRA-55 family data sets while they provide a good opportunity for a clear comparison owing to the articulate design. In order to better understand the differences of MSSWs, we further describe differences of daily geopotential height fields between JRA-55 and JRA-55C,





and relate them to the occurrence of MSSWs. We also explore strong wave forcings and associated stratospheric vortex responses for JRA-55 and JRA-55AMIP.

The rest of the paper is organized as follows. Section 2 explains the data and analysis methods used in this study. Section 3 surveys the climatology and variability in the extratropical stratosphere in the JRA-55 family data sets. Section 4 further examines JRA-55 and JRA-55C, and Section 5 examines JRA-55 and JRA-55AMIP. Finally, Section 6 provides summary and discussion.

## 2 Data, and analysis method

### 2.1 Data

This study makes use of daily averages for the three products of the JRA-55 family data sets. The horizontal resolution is $2.5^{\circ} \times 2.5^{\circ}$, with 37 levels up to 1 hPa. The full period is from 1958 to 2012, but we use a shorter period in some comparisons (see below). We mainly use the zonal mean zonal wind, poleward eddy heat flux by wave components of zonal wave numbers 1-3 (waves 1-3), and geopotential height. The eddy heat flux in the extratropical lower stratosphere (e.g., 40-90$^{\circ}$ N, 100 hPa) is used as a proxy for planetary wave forcing from the troposphere, since it is proportional to the vertical component of the Eliassen-Palm (EP) flux under the quasi-geostrophic scaling (Andrews et al., 1987).

We regard the standard JRA-55 reanalysis data (Kobayashi et al., 2015) as a good representation of the real world, and refer to it as STDD. A comparison of JRA-55C (referred to as CONV) to STDD elucidates effects of assimilation of satellite data, since the inclusion or exclusion of assimilation of satellite data is the only difference between the two (Kobayashi et al., 2014). We also compare JRA-55AMIP (referred to as AMIP) to STDD to examine model biases, since the AMIP data are obtained from an AMIP-type simulation using the same numerical weather prediction model without assimilation of any observational data.

For the comparison between STDD and CONV, we use the data from 1972/73 to 2012/13 when CONV is available. For the comparison between STDD and AMIP, we use the data from 1957/58 to 2012/13 (i.e., full period). We sort the data in time for both hemispheres so that they begin from June in each year and end in May in the next year to facilitate our main focus on NH winter when the NH extratropical stratosphere is dynamically active. For both NH and SH, we refer to each year from June to May by the year to which the month of January belongs.

### 2.2 Analysis methods

#### 2.2.1 Identification of MSSW onset dates

The method outlined by Charlton and Polvani (2007) is basically followed to identify MSSWs. This method identifies the onset date (denoted as lag= 0 day) of a MSSW as when the zonal mean zonal wind at 60$^{\circ}$ N, 10 hPa reverses from a westerly wind to an easterly wind. We focus on MSSWs during the winter period of December-January-February (DJF). In order to





identify two (or more) MSSWs in one season, the onset dates between two successive events must be separated more than 20 days, and the zonal wind must recover above 20 m s$^{-1}$ between them. The latter condition is added to ensure that the polar vortex is sufficiently re-established after the first event. The resultant onset dates of the MSSWs identified for STDD and CONV are shown in Table 1. The onset dates for STDD are identical to those in Butler et al. (2017), as far as the DJF MSSWs are concerned.

### 2.2.2 Calculation of polar vortex geometry

In order to characterize geometry of the polar vortex, centroid latitude (CL) and aspect ratio (AR) are calculated for the 10 hPa height according to Seviour et al. (2013). This method diagnoses where the center of the vortex is located (CL) and how stretched the vortex is (AR), where the vortex is defined as the region of the 10 hPa height lower than a threshold of Zb. The parameter Zb is taken as the climatological and zonal mean of the 10 hPa height at 60$^{\circ}$N in each product, and is different among the three products. The height fields are smoothed with a 5 day running mean in calculating CL and AR so as to filter out day-to-day fluctuations and capture dominant geometry features.

### 2.2.3 Calculation of RMSD of geopotential height fields

Differences in daily geopotential height fields, e.g., at 10 hPa, between STDD and CONV, are evaluated with the root mean square difference (RMSD) as follows:

$$\text{RMSD} = \left[ \frac{\sum_{i=1}^{n} w_i \{ Z_{CONV}(x_i) - Z_{STDD}(x_i) \}^2}{\sum_{i=1}^{n} w_i} \right]^{1/2}. \quad (1)$$

Here, $Z_{STDD}$ and $Z_{CONV}$ denote geopotential height fields at a level on a day of interest for STDD and CONV, respectively, $x_i$ denotes spatial grid points (longitude and latitude), and $w_i$ is cosine of latitude. The summations are taken for all extratropical grid points (indexed with i from 1 to n) poleward of 30$^{\circ}$N/S. Note that an arbitrary RMSD value is divided into contributions from the zonal and wave components as follows:

$$\text{RMSD}^2 = \text{RMSD}_{zonal}^2 + \text{RMSD}_{wave}^2. \quad (2)$$

Here, $\text{RMSD}_{wave}$ is calculated by applying Eq. (1) to wave fields of $Z_{STDD}$ and $Z_{CONV}$.

### 3 Survey of climatology and variability in the extratropical stratosphere during NH winter and SH spring

This section surveys the climatology (long-term mean) and variability in the extratropical stratosphere for NH winter and SH spring using the zonal mean zonal wind and 10 hPa geopotential height.

### 3.1 Climatology

Figure 1 shows the climatological zonal mean zonal wind in DJF and September-October-November (SON) for STDD, CONV, and AMIP. Color shades plot differences of CONV or AMIP from STDD that are judged to be statistically





significant according to Student t test (two-side test) at the 95 % level. When taking differences between STDD and CONV, the STDD data after the 1972/1973 season are used as stated in Sect. 2.1.

The climatological zonal wind is similar between STDD and CONV in a large part of the domain below the middle stratosphere for both seasons, although some differences are notable in the upper stratosphere. The differences are positive near the westerly jets in both hemispheres during the cold seasons, which indicate the stronger westerly jets for CONV. The SH easterly winds for DJF also have positive differences. Negative differences appear in NH subtropical and SH mid-latitudes for SON. Tropospheric wind differences are generally small in magnitude.

The wind differences of AMIP from STDD are roughly similar to those of CONV in spatial pattern, but are larger in vertical extent and magnitude in the stratosphere. AMIP also shows significant differences in the troposphere, although their magnitudes are smaller.

These differences in the polar night jet are consistently reflected in Fig. 2, which similarly shows maps of the climatological 10 hPa geopotential height. The polar vortex is very similar between STDD and CONV in strength and shape for both NH winter and SH spring (Fig. 2b,e). This feature corresponds to the absence of significant differences in the zonal wind at 10 hPa (Fig. 1b,e). On the other hand, AMIP simulates the vortex that is stronger than the STDD counterpart for both seasons (Fig. 2c,f), consistent with the positive wind differences around the polar night jet from the geostrophic wind relationship (Fig. 1c,f). Positive height differences are also notable in surrounding mid-latitudes.

Some of these results about the climatological zonal wind are consistently seen in Kobayashi et al. (2014) and Kobayashi and Iwasaki (2016). The latter study further claimed that the stronger polar vortex in CONV and AMIP reflects weaker wave propagation and driving.

## 3.2 Variability

Next, we examine variability in the extratropical stratosphere by looking at daily time series of the zonal mean zonal wind at $60^{\circ}$ N/S, 10 hPa (Fig. 3). The zonal wind at the grid points are used as a proxy for strength (and also flow direction) of the polar vortex for cold seasons in each hemisphere. The climatological seasonal cycle (long-term mean) and variability (standard deviation of interannual variability for each day) are also plotted in the figure. The climatology and standard deviation are smoothed in time so that they consist of low frequency components with periods longer than about 100 days.

The climatology and standard deviation of the zonal wind overlap between STDD and CONV for both NH and SH (Fig. 3b,e). A comparison of probability distribution functions (PDFs) of the daily zonal wind at $60^{\circ}$ N for DJF and at $60^{\circ}$ S for SON between STDD and CONV shows that they are very close to each other (Fig. 4b,e). It is difficult to notice frequency differences of easterly winds, or MSSWs in NH during DJF in Figs. 3b and 4b, which are further examined in Sect. 3.3. In contrast, one can see that STDD shows SH easterly winds in late September (MSSW in September 2002) whereas CONV underrepresents it as a minor SSW without a zonal wind reversal.

AMIP has the stronger climatological wind from mid-winter to spring, and somewhat smaller variability around January in NH. One also sees in Fig. 3a,c that zonal wind reversals during DJF are less frequent for AMIP than for STDD. These





features are reflected in PDFs of daily zonal wind data for STDD and AMIP (Fig. 4c). It is clear that the PDF for AMIP is biased toward the positive side, consistent with the stronger climatological westerly wind and less frequent zonal wind reversals. It is also notable that apart from the climatological difference, the daily zonal wind data for AMIP has smaller variability. Another PDF is drawn in thin solid line in Fig. 4c for zonal wind data for AMIP that are artificially decreased by the climatological wind difference between STDD and AMIP. Thus, this PDF has the same mean value as the STDD PDF, but is narrower with lower frequencies of extreme (both strong and weak) wind values.

The AMIP zonal wind data are also biased toward stronger vortex states in the SH extratropical stratosphere for SH winter and spring (Figs. 3f and 4f). The stronger climatological wind is notable from September to November. No MSSW (zonal wind reversal) is simulated from September to mid-October in AMIP.

## 3.3 Frequency and vortex geometry of MSSWs in NH

Figures 3 and 4 showed that the zonal wind variability in NH looks similar between STDD and CONV, suggesting that the occurrence of MSSWs is also similar. On the other hand, AMIP clearly lacks the zonal wind variability and easterly winds, suggestive of fewer MSSWs. In this subsection, we examine the frequency and also vortex geometry of NH MSSWs for the three products.

Table 1 lists the onset dates of DJF MSSWs identified for STDD and CONV. The frequency of DJF MSSWs for STDD is 30 events in the 56 seasons (53.6 % for each season) and 22 events in the 41 seasons after 1972/73 (53.7 %). CONV shows 19 events in the 41 seasons (46.3 %). A comparison of the MSSW onset dates between STDD and CONV in the 41 seasons shows that CONV reproduces most MSSWs in STDD as inferred from Fig. 3, but delays seven cases by one to four days. CONV also misses three cases, underrepresenting them as minor SSWs that do not accompany a zonal wind reversal. The latter feature is similar to the SH MSSW in September 2002, when CONV represents it as a minor SSW (Fig. 3). It is also noted that no opposite case exists where an onset date is represented earlier in CONV than in STDD or where a MSSW is represented only in CONV. The delayed and missed cases seem distributed randomly in time (year).

The frequency of DJF MSSWs for AMIP is 9 for the 56 seasons (16.1 %), which is much smaller than for STDD. The lower frequency of MSSWs in AMIP is consistent with the stronger westerly wind (Figs. 1-4).

The geometry of the polar vortex during the MSSWs for STDD is characterized in a scatter plot between CL and AR on lag= 0 day (Fig. 5a). One sees that the data points for STDD roughly form a linear distribution, with a correlation coefficient of +0.68. Data points located near the lower-left end correspond to MSSWs of low CL (e.g., vortex displacement MSSWs), and those near the opposite end correspond to MSSWs of high AR (e.g., vortex split MSSWs). Such a linear distribution was pointed out by Taguchi (2016). This distribution reflects that the zonal mean zonal wind at $60°$ N, 10 hPa representing the vortex strength tends to weaken as CL becomes low and/or AR becomes high (Fig. 5d).

CONV roughly reproduces a similar linear distribution of CL and AR on the MSSW onset dates. CONV tends to underestimate equatorward shift of the polar vortex for MSSWs located near the lower-left end, as CL for CONV is larger than the STDD counterpart. On the other hand, the differences in AR for MSSWs located near the opposite end vary: some

MSSWs for CONV overestimate AR compared to the STDD counterparts, whereas underestimations are also notable for others.

In addition to the lower frequency, the MSSWs for AMIP show a notable feature about the vortex geometry that all MSSWs have low CL and AR. No MSSW in AMIP shows high AR, e.g., over 2, on lag= 0 day. The absence of MSSWs of high AR is consistent with the scatter plot for all DJF data for AMIP (Fig. 5f): the data points for AMIP are biased toward high CL and low AR.

In the following, we further explore these differences in the frequency and vortex geometry of MSSWs. In Sect. 4, we relate the occurrence of MSSWs to general RMSD distributions of geopotential height fields between STDD and CONV. In Sect. 5, we examine strong planetary wave forcings and associated stratospheric vortex weakenings for STDD and AMIP.

## 4 Comparison of CONV to STDD

In Sect. 3, we showed that CONV is close to STDD in terms of the climatological zonal mean zonal wind and height in the NH extratropical stratosphere (Figs. 1 and 2). The daily time series and PDFs of the zonal wind also look similar between the two runs (Figs. 3 and 4), although the occurrence of MSSWs is slightly different. In this section, we describe general RMSD distributions of daily extratropical height fields between STDD and CONV at various levels, and relate them to the occurrence of MSSWs.

### 4.1 Climatological RMSD distributions

First, we examine climatological distributions of RMSD for NH and SH. Figure 6a,c plots the long-term means of RMSD as a function of time (season) and height. It is common between the two hemispheres and all seasons that RMSD increases with height as may be expected. In the NH upper stratosphere, RMSD has a semi-annual cycle, with two peaks in NH summer and winter. The winter peak is larger than the summer counterpart, and extends deeper down to the middle stratosphere such as 10 hPa. Such a semi-annual cycle is also notable in the SH upper stratosphere. The winter peak also extends deeper than the summer peak in SH as in NH.

The climatological height distributions of RMSD for NH winter and SH spring are extracted in Fig. 5b,d to emphasize interhemispheric differences during the dynamically active seasons. It confirms the increase in RMSD with height for both hemispheres. It also reveals that RMSD is larger in SH than in NH in a large part of the domain, except for the upper stratosphere. The SH has considerable magnitudes of RMSD even in the troposphere.

An examination of year-to-year changes in RMSD suggests a trend in SH, which is discussed in Sect. 6.2.

### 4.2 Case-to-case variability in RMSD, and MSSWs in NH

We further examine day-to-day variability in RMSD, particularly cases of extreme RMSD values, and compare them to the occurrence of MSSWs. Figure 7a,b shows daily time series of RMSD at 10 hPa in both NH and SH during dynamically



active seasons. PDFs of the RMSD values are also plotted in Fig. 6c,d: DJF and SON data are used for NH and SH, respectively. In addition to the climatological difference, typical minimum and maximum RMSD values as inferred from envelopes are also larger in SH than in NH. SH RMSD often attains large local maxima, e.g., over 400 m, whereas even in NH RMSD sometimes exhibits sharp peaks.

In order to examine specific geopotential height distributions, we identify cases of extremely large RMSD values for each of NH and SH as when RMSD attains local maxima over a threshold. The threshold is defined as the 95 percentile of all RMSD values during DJF in NH or SON in SH for the 41 seasons. The local maxima must be separated by more than 30 days when they are identified in the same season. RMSD values around the MSSW onset dates (from lag= -10 to +10 days) are excluded from this procedure, and are used separately.

Figure 8 presents scatter plots between contributions from the zonal and wave components to the total RMSD values for these cases. Note that the distance of each data point from the origin gives the total RMSD value (Eq. 2). The figure also includes the results on all MSSW onset dates identified in STDD. The data points use different colors and markers according to the zonal mean zonal wind at $60^{\circ}$ N/S, 10 hPa in STDD and its difference (CONV minus STDD) on the target day of each case (see the legends). The marker for each MSSW case accompanies a number (from 0 to 4) or a letter "M". The numbers

denote the time differences in the MSSW onset dates between STDD and CONV, and the letters "M" mean that these are missed in CONV (Table 1).

One sees in Fig. 8a that several data points that are located far from the origin and hence have largest total RMSE values are not associated with the MSSWs. There is one exceptional MSSW case accompanied by a number of 3. Some of the data points are located relatively close to the y-axis, implying large contributions from the wave component. The zonal mean

zonal wind in STDD is relatively strong (over 20 m s$^{-1}$) for these cases. The opposite also holds for other data points, when the zonal component makes large contributions and the zonal wind is weaker. Both overestimations and underestimations in the zonal wind by CONV occur for these cases.

Regarding the MSSWs, one may expect that the delayed and missed cases have larger RMSD values than the other MSSWs, but it turns out that this is not the case (see also Fig. 8c). Two of the delayed cases (with time differences of 2 and 4 days)

and all of the three missed cases appear in the intermediate data points near the coordinate point of (40, 40).

As for SH, a large part of the data points are located near the y-axis, implying an important role of the wave component. For these cases, STDD has strong westerly winds over 20 m s$^{-1}$, and both overestimations and underestimations in the zonal wind by CONV occur. The missed MSSW case (September 2002 MSSW in STDD) also occurs with a large contribution from the wave component.

Figure 9a-e plots 10 hPa height maps for the five largest RMSD values in NH selected from Fig. 8a. One sees large differences in places, which often occur near the edge of the polar vortex rather than near the center of the vortex. These differences do not have strong zonal symmetry but vary in the zonal direction, consistent with the large contributions from the wave component (Fig. 8a). These differences are reflected in different distributions of contour lines of 30000 m representative of the vortex edge, implying different shapes of the vortex between STDD and CONV.



The height differences during the four MSSWs (three missed cases and one delayed case, Fig. 9f-i) are generally smaller in magnitude. Both STDD and CONV show very similar vortex shapes in terms of the 30000 m contour lines. The difference fields have negative, albeit small in magnitude, values over polar latitudes, suggesting that the extratropical zonal wind is stronger in CONV. This is consistent with the fact that these MSSWs are missed or delayed in CONV.

One exception is the MSSW with the time difference of 3 days (Fig. 9j). In this case, the polar vortex similarly splits into two cyclones for both STDD and CONV. Nonetheless, the height differences have largely negative values in high latitudes. Figure 9k-o is similar, but for the SH cases of the five largest RMSD values (Fig. 8b). The differences in SH are generally larger in magnitude than the NH counterparts, and are dominated by wave 1 component. The dominance of wave 1 reflects that the location of the polar vortex (or distribution of the 30000 m contour) in CONV is notably deviated from the STDD

counterpart.

As for the SH MSSW, STDD shows a split of the vortex into two cyclones, whereas CONV shows that the vortex (30000 m contour line) shifts away from the South Pole and highly stretches, without a split (Fig. 8p). The height differences are comparable in magnitude to the preceding five cases. Negative values prevail in the height difference over polar latitudes, consistent with the underrepresentation of this case as a minor SSW in CONV.

## 15 5 Comparison of AMIP to STDD

We showed in Sect. 3 that the MSSW frequency for AMIP is much lower than that for STDD. In particular, AMIP lacks MSSWs of high AR. In order to better understand these differences, in this section we examine extreme planetary wave forcings (poleward eddy heat fluxes) and stratospheric vortex weakening responses to them. Since it is well known that MSSWs are a response of the polar vortex to anomalously strong planetary wave forcings from the troposphere (Matsuno,

1971; Limpasuvan et al., 2004), the differences may be explained in terms of either or both of wave forcing from the troposphere and vortex response in the stratosphere.

### 5.1 Lower MSSW frequency for AMIP

The first issue is the lower frequency of MSSWs in AMIP. One may hypothesize that it is due to a lower frequency of strong wave forcings and/or weaker vortex responses to such wave forcings.

In Fig. 10a, we examine how the poleward eddy heat flux in the extratropical lower stratosphere (40-90$^\circ$ N, 100 hPa) and zonal wind deceleration are related for all MSSWs in STDD and AMIP. The heat flux is averaged in time from lag= -20 to 0 day, and the zonal wind deceleration is also calculated for the 21 days. We focus on the 21 day period, since the correlation between the two quantities for the MSSWs in STDD maximizes when we average the heat flux for about 10 to 30 days (not shown). This is consistent with the fact that polar vortex strength is highly correlated to the heat flux when the latter is

averaged for a few weeks or longer (Polvani and Waugh, 2004). This method follows Taguchi (2017), who examined the Coupled Model Intercomparison Project Phase 5 (CMIP5) historical simulations with 30 models. It is common between



STDD and AMIP that as expected, the zonal wind decelerates more strongly when the wave forcing is stronger. The distribution for STDD has a correlation coefficient of -0.70.

Since this plot is based on the data around the MSSWs, it does not explain how/why the MSSWs are much fewer for AMIP. In order to obtain a clue for the difference in the MSSW frequency, we look at extreme planetary wave forcings by

identifying a maximum of the poleward heat flux (averaged for 21 days, from 20 day before to each day) of waves 1-3 for each winter season in STDD and AMIP. We also extract zonal wind decelerations associated with the maximum wave forcings (Fig. 10b).

The plot shows two features in addition to similar linear distributions between the wave forcing and wind deceleration for both STDD and AMIP as in the MSSWs. First, AMIP lacks relatively strong forcings. For example, STDD has ten samples

over a threshold of 25 K m s$^{-1}$, whereas AMIP has only three. Second, the zonal wind decelerations for wave forcings around 20 K m s$^{-1}$ seem stronger for STDD than for AMIP.

The second point is further examined in a composite analysis with respect to the days of the maximum heat flux that ranges from the 25 percentile (17.4 K m s$^{-1}$) to the 75 percentile (23.9 K m s$^{-1}$) of all maximum heat flux values in STDD (Fig. 11). The range limitation intends to subsample wave forcings of similar magnitudes for both STDD and AMIP.

As a result, the composite daily heat flux increases from lag≈ -20 to 0 day for both STDD and AMIP (Fig. 11c). The lag= 0 day here denotes when the 21 day mean heat flux maximizes. This feature is quite similar between the two products, and the 21 day means from lag= -20 to 0 day are not significantly different at the 95 % level. The composite residual mean poleward wind and wave driving (EP flux divergence/convergence) are also similar between the two (Fig. 11b,d), and their 21 day means are not significantly different at the 95 % level. In contrast, the composite zonal wind evolution is different, as the

zonal wind decelerates more strongly for STDD. It is around 40 m s$^{-1}$ before lag≈ -20 days in both STDD and AMIP, before decreasing to lag≈ 0 day. The wind deceleration from lag= -20 to 0 day is 28.5 m s$^{-1}$ for STDD and 17.6 m s$^{-1}$ for AMIP, and this difference is judged to be significantly different at the 95 % level. This feature is consistent with Fig. 10b.

Thus, these results suggest that the lower frequency of MSSWs for AMIP is contributed by the lower frequency of extreme wave forcings, and the weaker vortex response in the stratosphere. These differences may reflect model biases in AMIP (or

effects of data assimilation in STDD), which will be discussed in Sect. 6.

## 5.2 Lack of MSSWs of high AR in AMIP

The second issue is the lack of MSSWs of high AR in AMIP. Again, this may be due to a lack of appropriate wave forcings leading to such MSSWs, and/or a difference of vortex responses to such forcings. Charlton and Polvani (2007) showed that vortex split MSSWs, which are characterized by high AR, are predominantly associated with amplified wave 2 activity.

Such a relationship between planetary wave forcings and MSSWs of high AR is seen in Fig. 12a, which is a scatter plot between wave 1 and wave 2 heat flux (averaged from lag= -20 to 0 day) for the MSSWs in STDD (crosses) and AMIP (circles). The data points are colored according to AR values on lag= 0 day (see the colorbar). Panel (c) is similar, but looks for a maximum AR value from 10 days before to 10 days after each MSSW onset date, and use the day of the AR maximum



as the key in calculating the 21 day mean heat flux. This treatment is included to capture increases in AR that do not always occur on the MSSW onset dates but sometimes occur around them. The plots show that some MSSWs of high AR values over 4 occur when wave 2 forcing is strong over ≈10 K m s⁻¹ and wave 1 forcing is weak below ≈5 K m s⁻¹. There are other MSSWs of high AR values that are accompanied by large wave 1 forcing over ≈10 K m s⁻¹ even when wave 2 forcing is relatively weak. The inclusion of the time lead or lag in identifying the AR maxima contributes to the increases in AR.

Since these plots, again, focus on the MSSWs and do not reveal how/why AMIP lacks MSSWs of high AR, in Fig. 12b we examine similar pictures for the maximum of the 21 day mean heat flux of waves 1-3 for each winter identified in Sect. 5.1. In Fig. 12d, we further allow time lead or lag for identifying AR maxima as in the MSSW case. A comparison between STDD and AMIP shows that the distributions of the data points roughly overlap between the two. Nonetheless, extreme AR values over 4 occur only for STDD. These suggest that a different vortex response in the stratosphere to strong wave forcings from the troposphere plays an important role in explaining the lack of MSSWs of high AR in AMIP. The lack of extreme wave forcings for AMIP seen in Fig. 10b is not conspicuous in Fig. 12b,d, but can be confirmed when taking the sum of the wave 1 and wave 2 contributions (not shown).

It is noted that the climatological, or background, difference in the 10 hPa height will play a role in explaining the different vortex response. The climatological difference is negative and larger in magnitude as going toward polar latitudes (Fig. 2). Then, it is likely to follow that when the polar vortex in STDD largely stretches for a high AR value in response to a wave forcing, it will be more difficult for the deeper AMIP vortex to exhibit similar stretching even for the same forcing. This idea is related to Seviour et al. (2016), who pointed out a close connection between the frequency of vortex split MSSWs and climatological AR for 13 CMIP5 models.

## 6 Summary and discussion

### 6.1 Summary

This study has compared large-scale dynamical variability in the NH and SH extratropical stratosphere, such as MSSWs, among the JRA-55 family data sets. In spite of the importance of dynamical variability in the stratosphere, such an aspect was relatively unexplored in the JRA-55 family data sets and also other reanalyses. This study owes the comparison to the articulate design of the JRA-55 family data sets.

First, a survey of the climatological states confirms the stronger polar night jet in both hemispheres for CONV and AMIP than for STDD. This difference is more notable for AMIP. A comparison of MSSWs reveals that CONV reproduces a large part of MSSWs identified in STDD. However, CONV delays several cases by one to four days, and also misses three cases underrepresenting them as minor SSWs. The SH MSSW in September 2002 is also underrepresented as a minor SSW in CONV. AMIP lacks MSSWs, especially those of high AR values. These differences of CONV could be understood by the bias of the numerical weather prediction model (as seen in AMIP) and the paucity of data assimilation as hypothesized by Kobayashi et al. (2014).



Next, we examined daily RMSD distributions of geopotential height fields between STDD and CONV, and relate them to the occurrence of MSSWs. In spite of the slight climatological differences in the zonal mean zonal wind and 10 hPa geopotential height, the examination shows that CONV is sometimes significantly different from STDD in terms of daily height fields. The differences are large especially in the upper stratosphere in both hemispheres during dynamically active seasons, with larger values in SH. The RMSD values in NH winter sometimes have sharp peaks, apart from the MSSWs. The RMSD peaks include contributions from the wave component that are larger than or comparable to those from the zonal component. RMSD values for some of the delayed MSSWs and all of the missed MSSWs are smaller than such peaks. In such MSSW cases, CONV overestimates the zonal mean zonal wind in the extratropical stratosphere by definition, and both of the zonal and wave components make important contributions.

Furthermore, we examined strong planetary wave forcings and associated zonal wind decelerations (polar vortex weakenings) in STDD and AMIP. It turns out that in AMIP, extreme wave forcings are fewer and zonal wind decelerations to such wave forcings are weaker. These features are commonly seen in some of the CMIP5 historical simulations (Taguchi 2017). These model biases may be reflected in CONV, where they contribute to the delay and miss of several MSSWs. These two factors are also suggested to be relevant to the lack of MSSWs of high AR values in AMIP.

## 6.2 Discussion

Finally, we discuss several aspects unanswered in this study.

This study reveals large RMSD values for stratospheric height fields between STDD and CONV (Figs. 7-9), but does not fully answer how/why such differences occur. It does not seem simple to understand what patterns such large differences take and when they occur. They should be explained by the effects of the assimilation of satellite data, but future study is needed to understand the effects more specifically.

A further examination of year-to-year variations in RMSD at several levels suggests a decreasing trend in SH (Fig. 13). The SH decreasing trend is statistically significant at the 95 % level for all four levels. Such a trend is absent or much weaker in NH. This suggests that the impacts of assimilation of satellite data decrease over the decadal time scale especially in SH. Further details and mechanisms will be worthy of investigation.

The differences between STDD and AMIP, which are relevant to model biases, are characterized by the stronger climatological polar vortex and lack of MSSWs for AMIP. The lack of MSSWs in AMIP is not fully explained by the climatological difference, although it is consistent. This suggests that some additional factors or processes, such as tropospheric planetary wave forcing and stratospheric vortex response, play roles.

Detailed analyses of these aspects will be therefore useful. A decomposition technique of eddy heat flux (e.g., Nishii et al., 2009; Fletcher and Kushner, 2011) could be applied so as to better understand the lack of extreme wave forcings from the troposphere in AMIP. A zonal momentum budget analysis as in Martineau et al. (2016) could be also conducted to diagnose stratospheric vortex responses to strong wave forcings in the three products, including how/why several MSSWs are delayed or missed in CONV compared to STDD.





**Acknowledgements**

The author thanks the Japan Meteorological Agency for making the JRA-55 family data sets available. The data were obtained from the Research Data Archive at the National Center for Atmospheric Research, Computational and Information Systems Laboratory. This study is supported by the Grant-in-Aid for Scientific Research (S) 2422401101 and (C) 15K05286.

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



**Table 1: Onset dates (in the dd-month-yyyy format) of DJF MSSWs for STDD and CONV, with differences of CONV from STDD. Grey region indicates the period when CONV is unavailable. The letters "M" mean that the MSSWs identified in STDD are missed in CONV.**

| STDD | CONV | DIFF |
|------|------|------|
| 30 Jan 1958 | | |
| 17 Jan 1960 | | |
| 30 Jan 1963 | | |
| 18 Dec 1965 | | |
| 23 Feb 1966 | | |
| 07 Jan 1968 | | |
| 02 Jan 1970 | | |
| 18 Jan 1971 | | |
| 31 Jan 1973 | 01 Feb 1973 | 1 |
| 09 Jan 1977 | 13 Jan 1977 | 4 |
| 22 Feb 1979 | 22 Feb 1979 | 0 |
| 29 Feb 1980 | 29 Feb 1980 | 0 |
| 06 Feb 1981 | M | — |
| 04 Dec 1981 | M | — |
| 24 Feb 1984 | 24 Feb 1984 | 0 |
| 01 Jan 1985 | 04 Jan 1985 | 3 |
| 23 Jan 1987 | 23 Jan 1987 | 0 |
| 08 Dec 1987 | 08 Dec 1987 | 0 |
| 21 Feb 1989 | 21 Feb 1989 | 0 |
| 15 Dec 1998 | 16 Dec 1998 | 1 |
| 26 Feb 1999 | 26 Feb 1999 | 0 |
| 11 Feb 2001 | 11 Feb 2001 | 0 |
| 31 Dec 2001 | M | — |
| 18 Jan 2003 | 18 Jan 2003 | 0 |
| 05 Jan 2004 | 07 Jan 2004 | 2 |
| 21 Jan 2006 | 22 Jan 2006 | 1 |
| 24 Feb 2007 | 24 Feb 2007 | 0 |
| 22 Feb 2008 | 22 Feb 2008 | 0 |
| 24 Jan 2009 | 25 Jan 2009 | 1 |
| 09 Feb 2010 | 09 Feb 2010 | 0 |





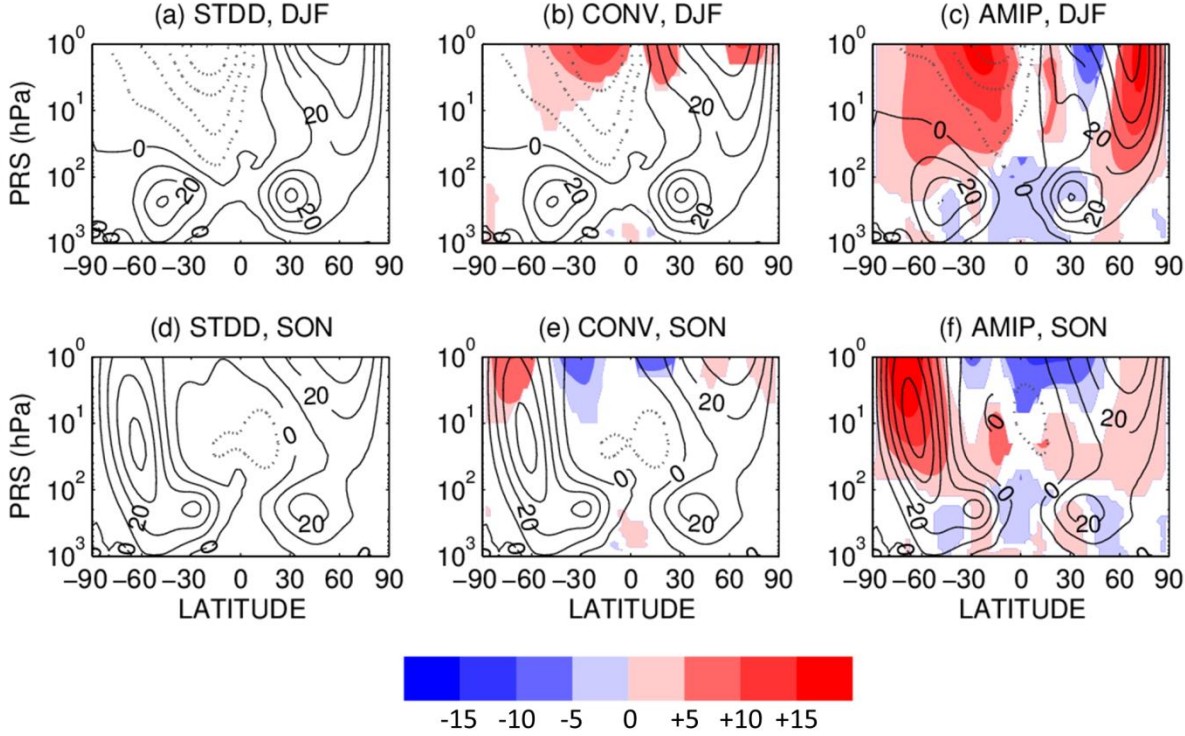

**Figure 1: The climatological zonal mean zonal wind in the JRA-55 family data sets for (a-c) DJF and (d-f) SON in black contours.
Panels (a,d) are for STDD, (b,e) for CONV, and (c,f) for AMIP. Contour interval is 10 m/s. Panels (b,c,e,f) also plot differences
from STDD by color shades (see the colorbar) that are statistically significant at the 95 % level. When taking differences of CONV
from STDD, the STDD data after the 1972/1973 season are used.**



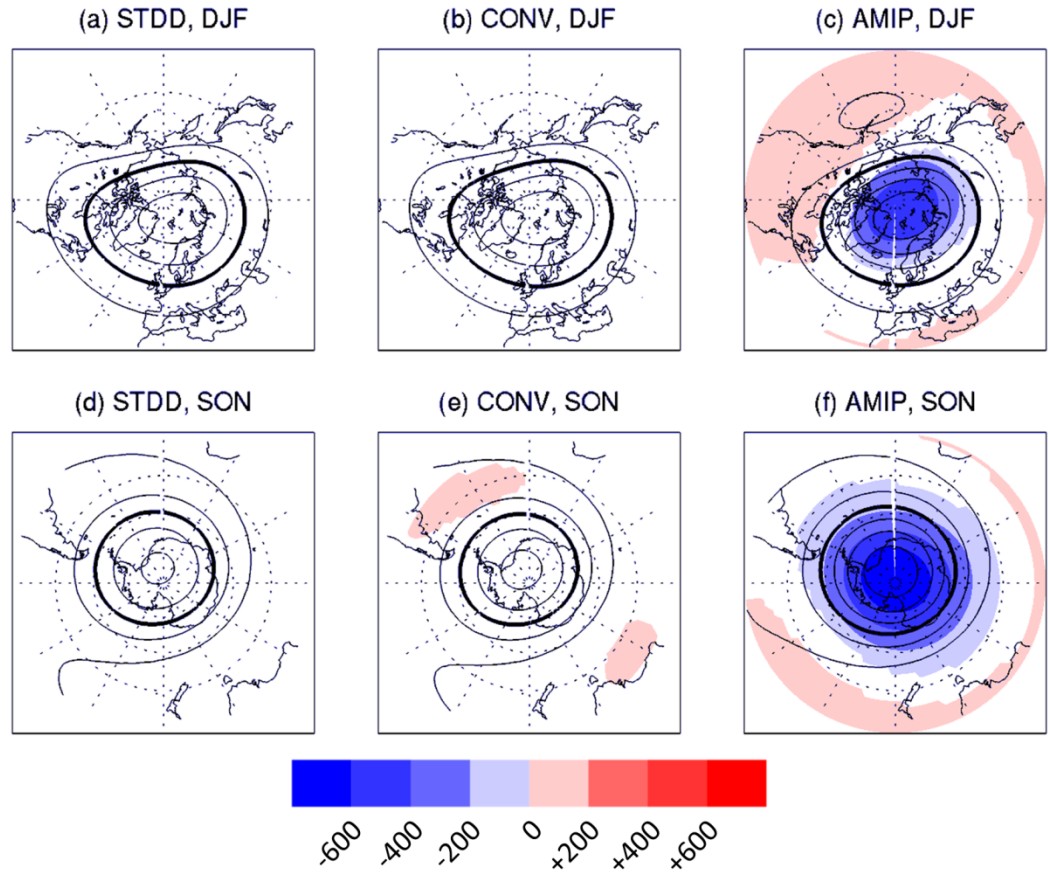

**Figure 2: Same as in Fig. 1, but for the climatological mean 10 hPa height. Contour interval is 500 m. Thick contours denote 30000 m.**



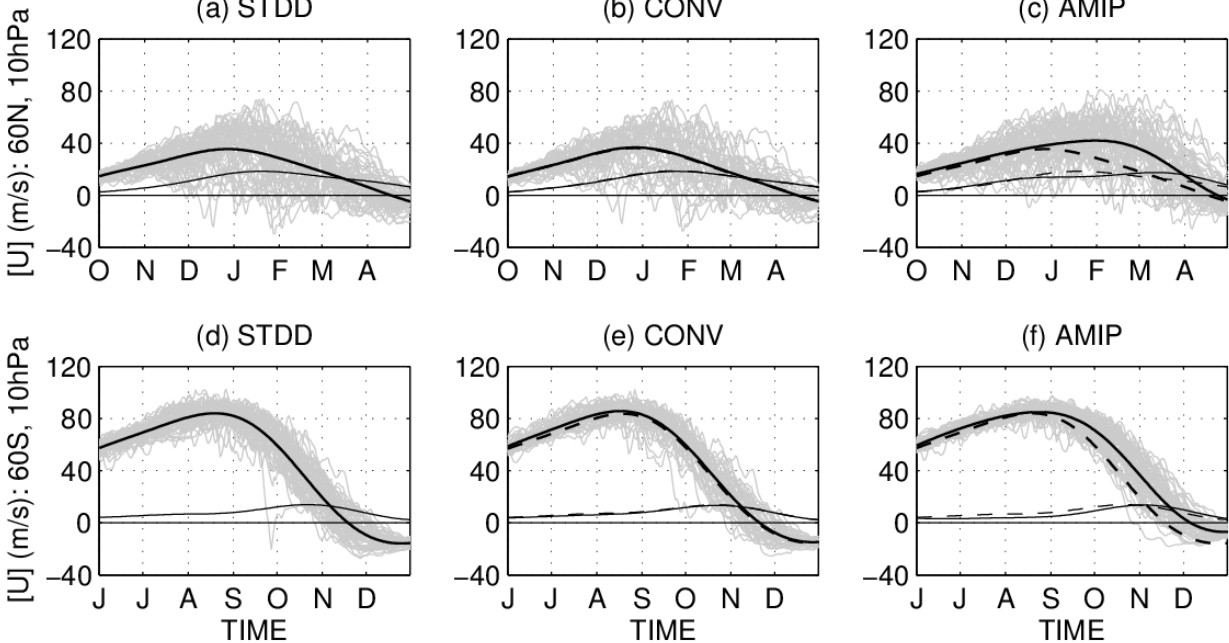

**Figure 3:** Daily time series of the zonal mean zonal wind [U] at 10 hPa: (a-c) 60° N, and (d-f) 60° S. The square brackets denote the zonal mean. Panels (a,d) are for STDD, (b,e) for CONV, and (e,f) for AMIP. Solid lines in each panel denote the climatological seasonal cycle, and standard deviation of interannual variability for each day. Broken lines in (b,c,e,f) denote the results from STDD. The STDD data after the 1972/1973 season are used in (b,e). Month labels are placed at the first day of each month.



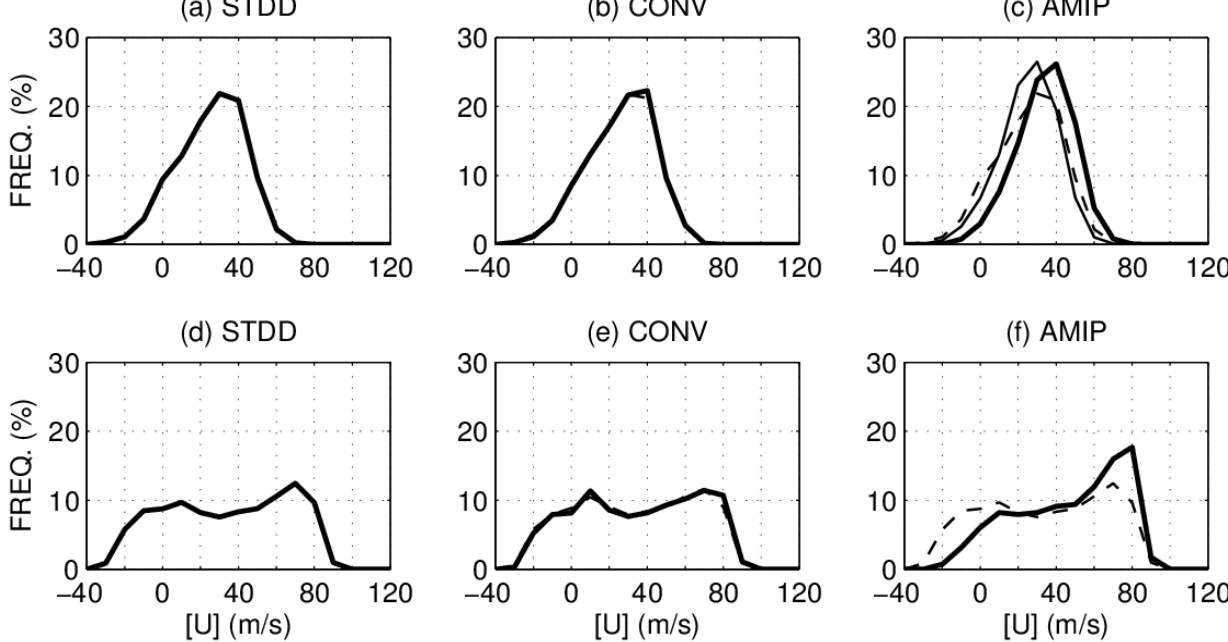

**Figure 4:** PDFs of the daily zonal mean zonal wind at 60° N, 10 hPa for DJF (a-c) and at 60° S, 10 hPa for SON (d-f) in thick solid lines. Panels (a,d) are for STDD, (b,e) for CONV, and (e,f) for AMIP. Broken lines in (b,c,e,f) denote the results from STDD. The STDD data after the 1972/1973 season are used in (b,e). In panel (c), an additional PDF is drawn in thin solid line for the AMIP zonal wind data that are artificially decreased by the climatological difference from STDD, so that the decreased wind data have the same mean value as in STDD.




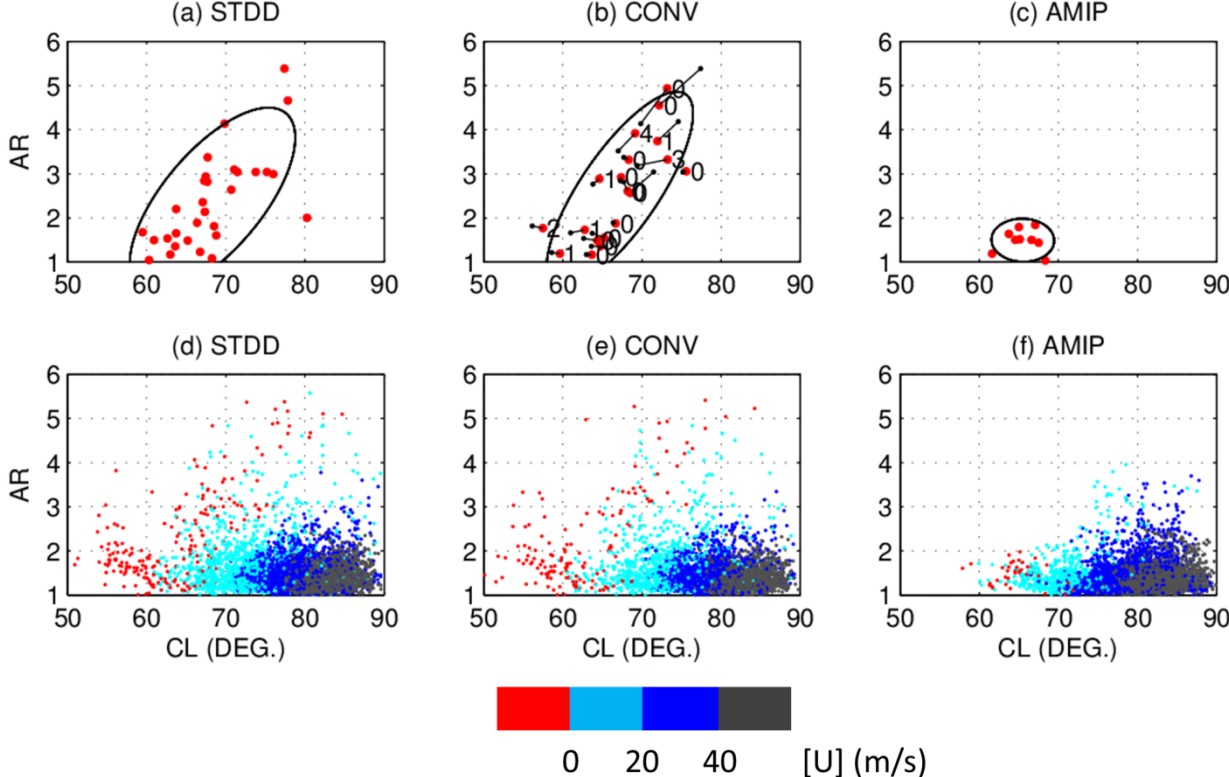

**Figure 5: (a-c) Scatter plots between CL and AR of the 10 hPa height on the onset dates of the NH MSSWs: (a) STDD, (b) CONV, and (c) AMIP. Ellipses denote representative distributions of the data points extracted by an empirical orthogonal function analysis. Each data point in (b) is connected to the corresponding case in STDD, and accompanied by a number denoting the time difference in the onset dates (Table 1). Panels (d-f) plot results for all DJF days available to each product. Each data point is colored by the zonal mean zonal wind at 60° N, 10 hPa (see the colorbar).**




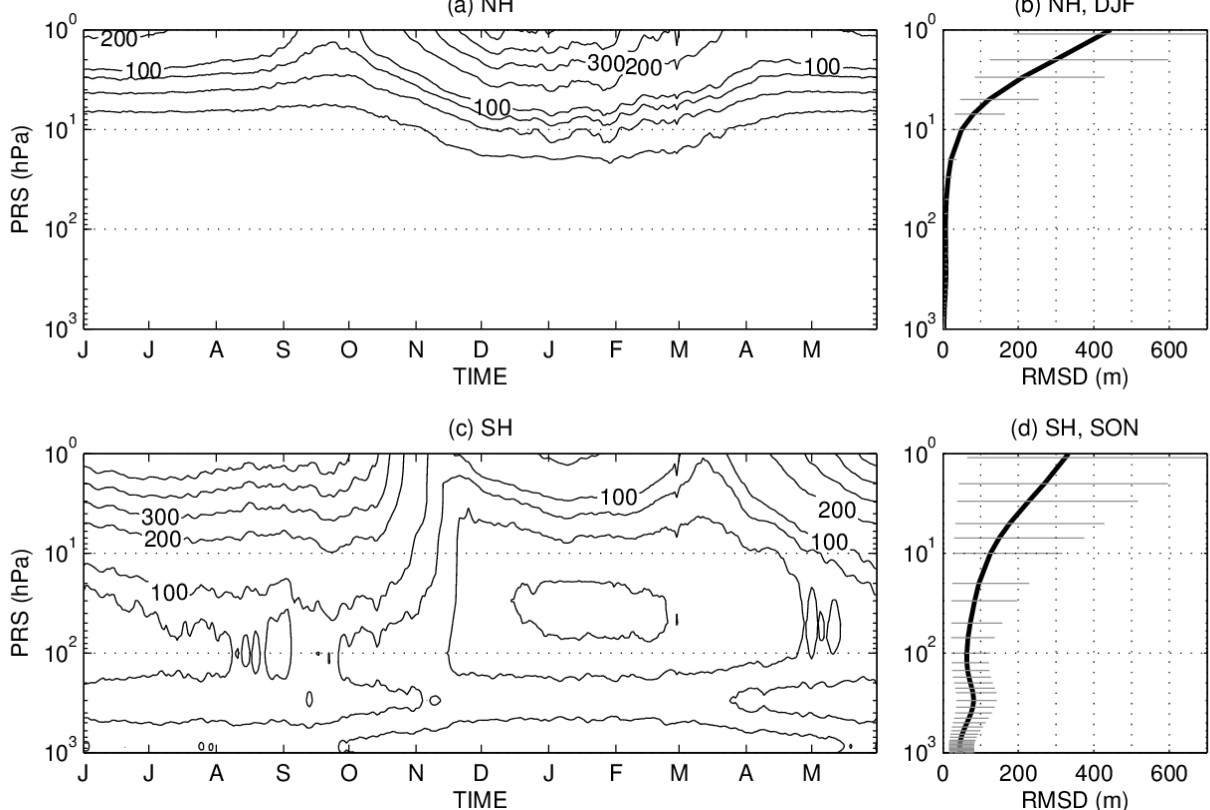

**Figure 6:** (a,c) Time-height sections of the climatology of RMSD: (a) NH, and (c) SH. Contour interval is 100 m, with additional contours at 25, 50, and 75 m. Panels (b,d) plot seasonal means as a function of height: (b) NH for DJF, and (d) SH for SON. Horizontal lines denote 5 and 95 percentiles.





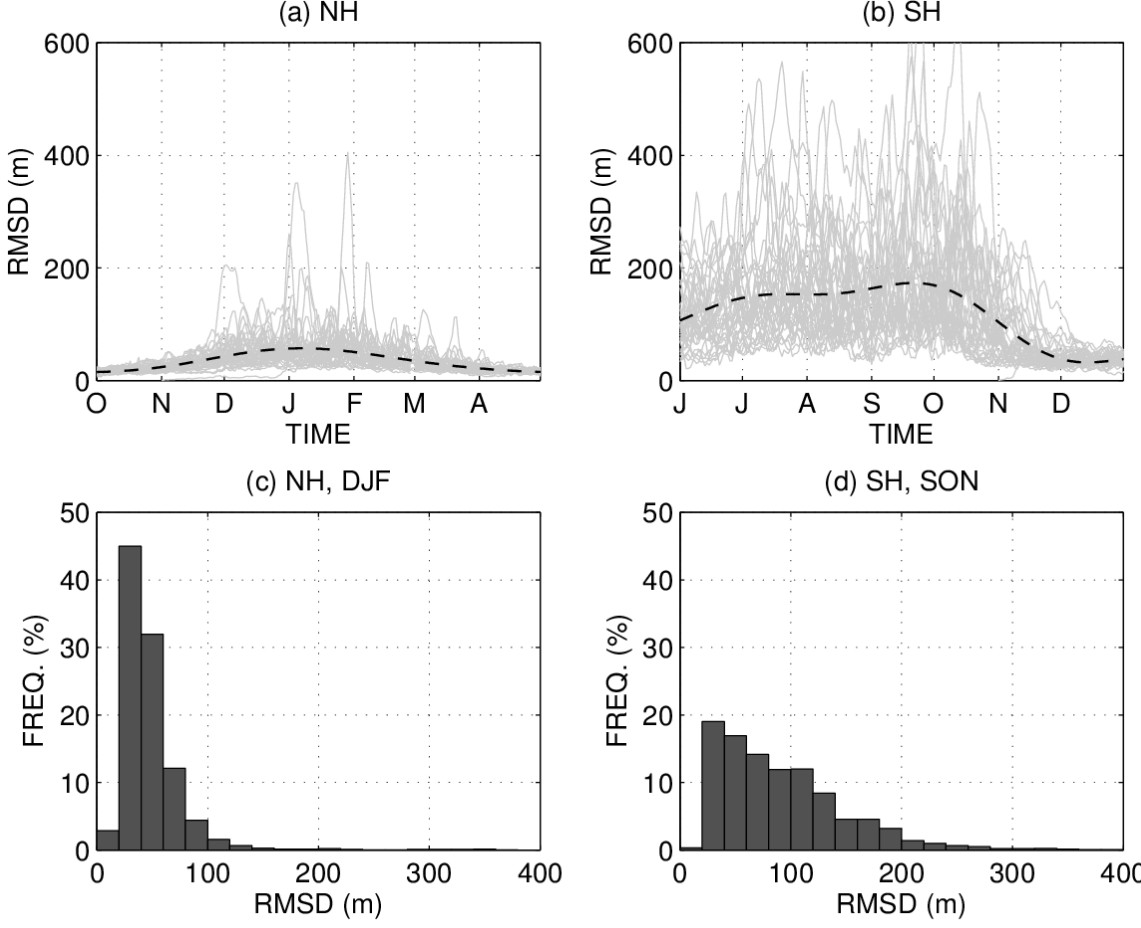

**Figure 7: (a,b) Daily time series of RMSD at 10 hPa: (a) NH, and (b) SH. Broken lines denote the climatological seasonal cycle. Panels (c,d) show PDFs of the RMSD values for DJF in NH and for SON in SH.**





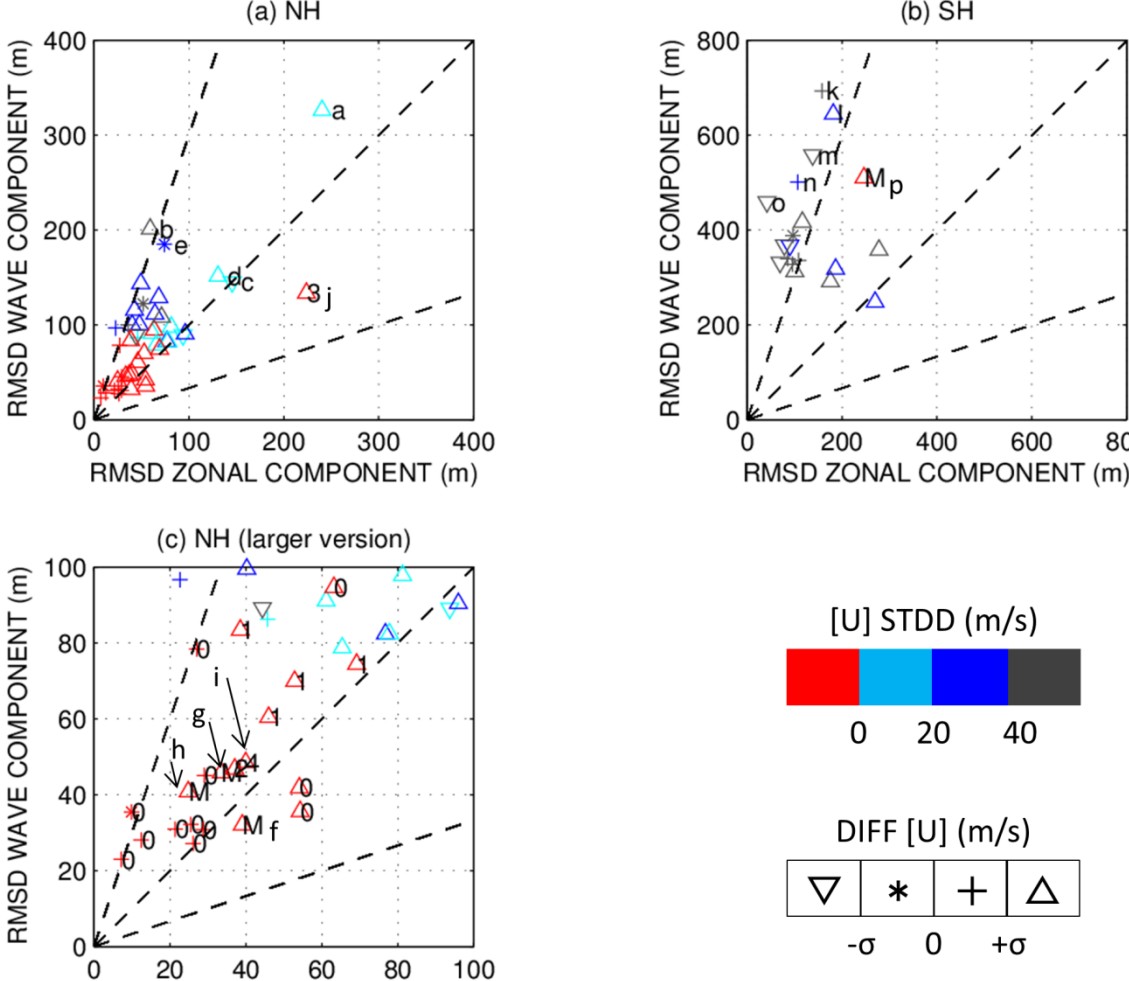

**Figure 8: (a,b) Scatter plots between zonal and wave contributions to total RMSD for large RMSD peaks exceeding the 95 percentile of all RMSD values in STDD: (a) NH for DJF, and (b) SH for SON. Results for all MSSW onset dates in STDD are also shown. Panel (c) is a larger version of (a). The data points are plotted in different colors and markers, which, respectively, denote the zonal mean zonal wind in STDD and its difference between STDD and CONV at 60° N/S, 10 hPa on the target dates (see the colorbar). The σ denotes the standard deviation of all zonal wind difference data between STDD and CONV: 1.9 m s⁻¹ for DJF in NH, and 3.7 m s⁻¹ for SON in SH. Letters a-e in (a) and k-o in (b) denote cases used in Fig. 9. Each data point for the MSSWs accompanies a number from 0 to 4 or a letter M. The numbers denote the time differences in the onset date, or the letters M mean that the MSSWs in STDD are missed in CONV (Table 1). The MSSW data points are further related to f-j and p, which denote the panels in Fig. 9. Broken lines denote y=x/3, y=x, and y=3x.**





**Figure 9: (Black contours in a-e and k-o)** Maps of the 10 hPa height in STDD for the five largest RMSD values: (a-e) NH, and (k-o) SH. These labels are the same as in Fig. 8a,b. The target dates are also denoted in the ddmmyyyy format. Contour interval is 500 m, and thick black contours denote 30000 m in STDD. Magenta contours denote 30000 m for the 10 hPa height in CONV on the same dates. Color shades denote differences of CONV from STDD (see the colorbar). Panels (f-j) are similar, but for the three missed MSSW cases and the two delayed cases in NH. These labels are also denoted in Fig. 8a,c. Panel (p) is similar, but for the 2002 MSSW in SH (see Fig. 8b).





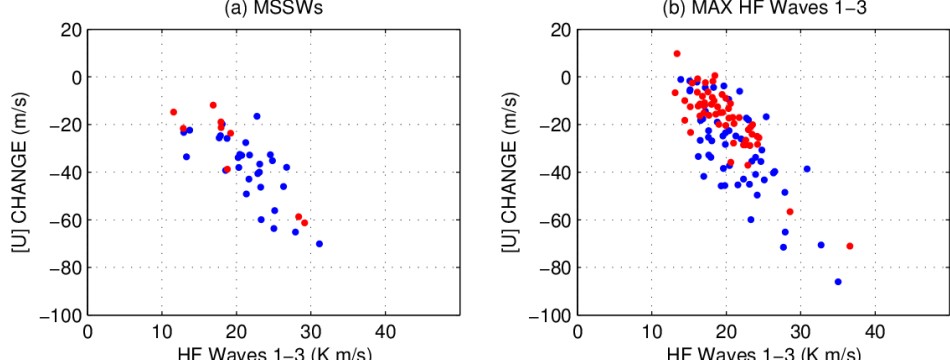

**Figure 10: (a) Scatter plot between 21 day mean heat flux (lag= -20 to 0 day) of waves 1-3, and associated zonal wind deceleration for the 21 days for all MSSWs in STDD (blue) and AMIP (red). Panel (b) is similar, but uses the maximum of the 21 day mean heat flux of waves 1-3 in each DJF season as the key.**





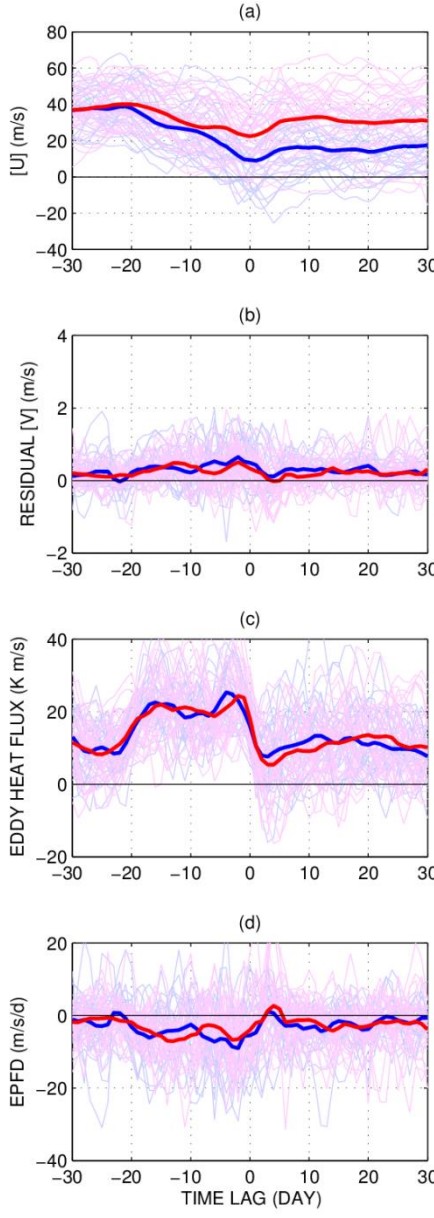

**Figure 11: Composite time series with respect to the maximum of the 21 day mean heat flux of wave 1-3 in each DJF season for STDD (blue) and AMIP (red): (a) zonal mean zonal wind at 60° N, 10 hPa, (b) residual mean meridional wind at 60° N, 10hPa, (c) heat flux of waves 1-3 at 40-90° N, 100 hPa, and (d) EP flux divergence/convergence of waves 1-3 at 60° N, 10 hPa. Thick lines denote composite results, and thin lines denote results for each case. The lag= 0 day denotes when the 21 day mean heat flux maximizes.**



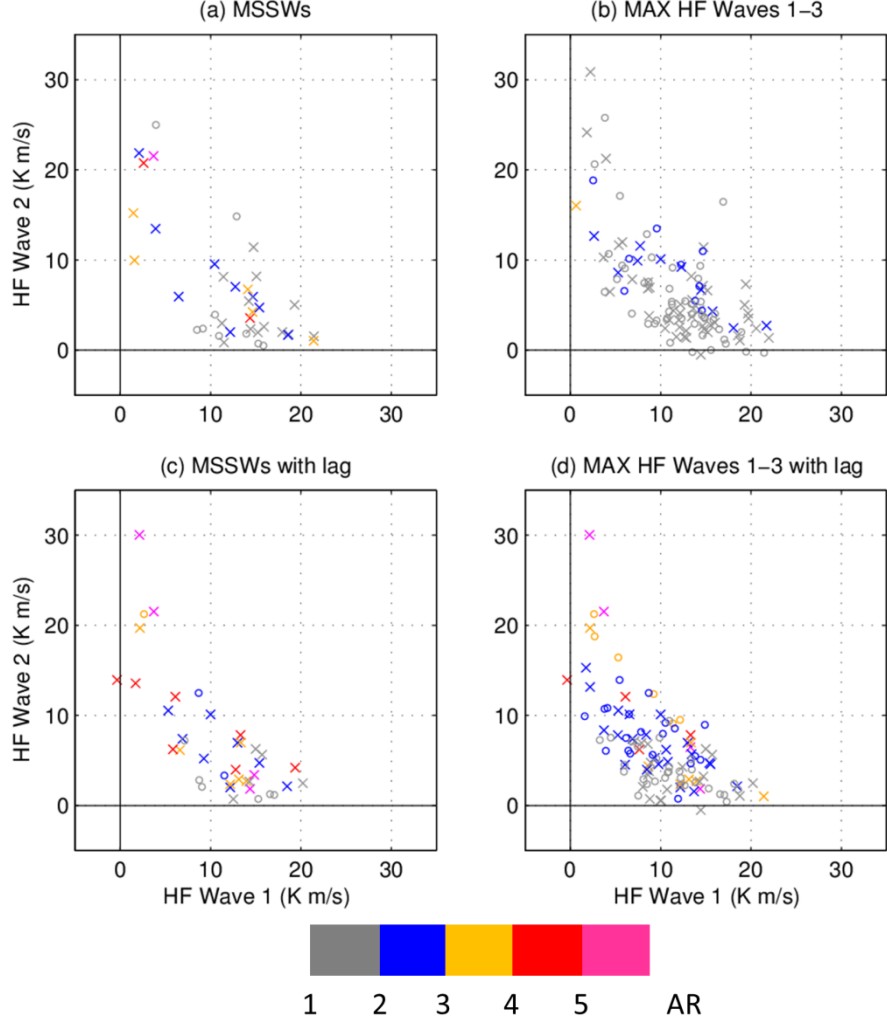

**Figure 12: (a) Scatter plot between wave 1 heat flux and wave 2 heat flux (both averaged from lag= -20 to 0 day in 40-90° N, 100 hPa) for all MSSWs in STDD (crosses) and AMIP (circles). Each data point is colored according to the AR value on the onset dates (see the colorbar). Panel (c) is similar, but looks for the AR maximum between 10 days before and 10 days after each MSSW onset date, and uses the day as the key. Panels (b,d) are similar, but use the maximum of 21 day mean heat flux of waves 1-3 (40-90° N, 100 hPa) in each DJF season instead of the MSSWs.**





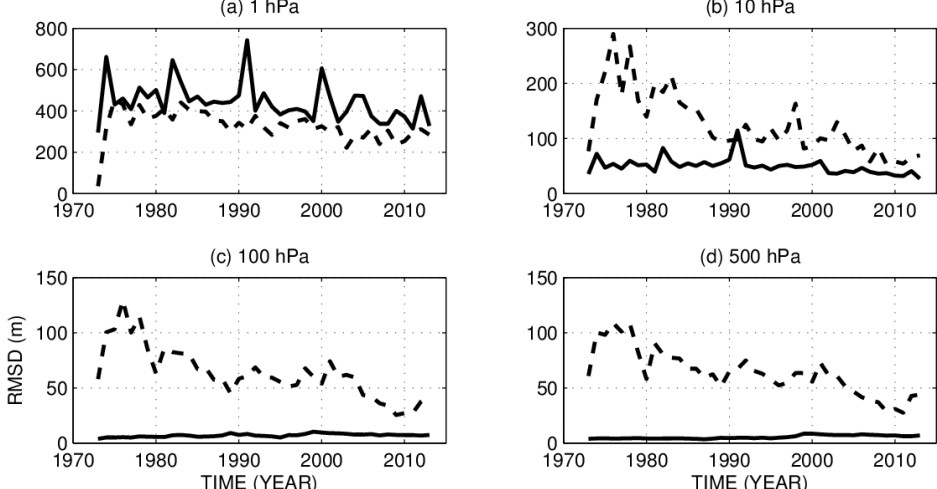

**Figure 13: Year-to-year variations of RMSD at four levels as indicated. Solid lines denote results in NH for DJF, and broken lines denote results in SH for SON.**