# Peer review of "Comparison of large-scale dynamical variability in the extratropical stratosphere among the JRA-55 family data sets"

_Atmospheric Chemistry and Physics, 2017_

## Referee Comment (RC1) · Anonymous Referee #1 · 30 May 2017

General Comments

This paper compares extratropical dynamic variability between three JRA-55 data sets: the standard reanalysis product (JRA-55), a product which assimilates all observations except for satellite data (JRA-55C), and a product which uses the same NWP model without any assimilation of observations (JRA-55AMIP). Without satellite observations, JRA-55C misses or delays several major sudden stratospheric warmings (MSSWs), and with no data assimilation, JRA-55AMIP has a much lower MSSW frequency. The author explores these differences in detail and with scientifically sound methods. I think this paper is appropriate for ACP and presents interesting results, and I would recommend for publication, after addressing my minor concerns below.

[Figure]

One general but minor comment I have is that the use of the STDD, CONV, and AMIP acronyms was not particularly useful and in fact made things more muddled. I would recommend just sticking with JRA-55, JRA-55C, and JRA-55AMIP. If the author thinks the acronyms add something, then the acronyms should at least be used earlier in the abstract/introduction so that readers become used to seeing them prior to section 2.

The other general comment I have is that I appreciated the creativity the author used to make the plots, some of which were very useful ways of visualizing the results; but Figure 8 went a bit overboard, with different symbols, colors, letters and numbers. Some of these attributes may be useful, like the colors; but the different symbols didn't seem to point to anything of particular interest. Right now the text reads (Page 8, Line 21-22 and Line 27-28): "Both overestimations and underestimations in the zonal wind by CONV occur for these cases", which indicates that these symbols are not adding much to our understanding and making the plot more difficult to read. The author could just make a note in the text that they also considered biases in the CONV winds relative to the STDD winds but didn't notice any systematic relationship.

Specific Comments

Page 1, Line 9, 19: It is not entirely clear what "conventional" observations include until section 2. It would be better to directly state here that this means everything except satellite data.

Page 1, Line 13: Readers unfamiliar with vortex geometry diagnostics may be unfamiliar with "aspect ratio" of the polar vortex; might be good to briefly relate this to displacements/splits here to be clear. Also in section 2.2.2.

Page 2, Line 17: Here, is "This study" referring to S-RIP, or to the current study?

Page 3, Line 21: prior to 1979, how different are the CONV and STDD products?

Page 5, Line 9-10: Might mention that the tropospheric jets seem shifted, which may (or may not) be consistent with biases in the stratosphere.

Page 6, Lines 32-33, Figure 5: Maybe I'm just confused about what exactly this plot is showing, but I don't understand why in panel (b), the black dots representing STDD don't seem to match the location of the STDD dots in panel (a). This makes lines 32-33 also confusing, as the statement does seem accurate based on the red/black dots in panel (b) but does not seem accurate when comparing panel (b) to panel (a).

Page 7, Line 5-6: Might mention that this implies few splits in AMIP.

Page 8, Line 19-20: the closeness to the y-axis may indicate a large contribution from the wave component, but it terms of proximity to the y-axis, there seems to be many "red"/weak wind points as well as strong ones. Might need to rephrase to clarify that for large values of the y-component, the zonal wind is stronger.

Page 9, Line 5: Just to clarify, in Figure 9 are you comparing the products using dates from STDD for both, or the central dates of each product?

Page 10, Line 12-13: Just to clarify, in Figure 11 are the grey lines all the cases between 25-75th percentile? Also, is the number of cases in your stated 25-75th percentile range similar for AMIP and STDD? It looks somewhat similar in Figure 10 but might be worth mentioning. Also you could consider drawing in light dashed lines the 25th and 75th percentile lines in Figure 10, as a way to clarify what you are plotting in Figure 11.

Page 12, Line 10-12: Is there a difference in latitude of maximum heat flux/EPFD forcing, or a difference in the maximum winds of the climatological polar jet in these data sets (or the edge of the vortex)? Could that also have an influence?

Page 15, Table 1 caption: make clear that "differences" are the number of days between onset dates.

Page 16, Figure 1: It wasn't entirely clear from this figure whether values between -5 to +5 are not significant (since they still are shaded)- could this be more clear? Or are any shaded values significant? I guess it's confusing since the colorbar doesn't have any white, non-significant level.

Page 18-19, Figures 3-4: I think these could be made more clear by staying consistent across panels and always making STDD line dashed, even in panel (a),(d). Also state in the caption what the non-bold and bold lines refer to.

Page 19, Figure 4: I would add in DJF to the plot titles for the top row and SON to the plot titles for the bottom row.

Page 26, Figure 11: by residual [v], do you mean the TEM term v-bar star? Or what is meant by residual [v]? (and page 10, line 17).

Technical Corrections

Page 1, Line 18: change "It shows" to "We find"

Page 1, Line 21: "vital" is maybe not quite the right word here

Page 1, Line 26: "Some of weak conditions"→ remove "of"

Page 2, Line 5: "the middle atmosphere science"; remove "the", change to middle atmosphere dynamics

Page 2, Line 10: Missing "with" after "associated"

Page 2, Line 11: "metrological" should be "meteorological"

Page 2, Line 13: I would change "go along with" to "are part of"

Page 2, Line 33: I'm not sure what is meant by "articulate design". Maybe "meticulous" would work better? Same with Page 11, Line 25.

Page 5, Line 22: change "at the grid points are" to "at this location is"

Page 5, Line 32: delete "the", change "wind" to "winds"

Page 7, Line 23: should be Fig 6b,d.

Page 8, Line 3: change "whereas even in" to "and even" (remove "in")

[Figure]

Page 8, Line 10: change to "Figure 8 presents scatter plots of the zonal and wave components contributing to the total RMSD values for these cases."

Page 8, Line 12: Change "on" to "of"

Page 9, Line 20-21: change to "explained in terms of wave forcing from the troposphere and/or the vortex response in the stratosphere."

Page 10, Line 13: change to "25th" and "75th"

Page 10, Line 16: might say "This feature is quite similar between the two products by construction" since you have chosen the range of heat flux values to be similar.

Page 10, Line 23: not sure "contributed by" is the right phrase here. Maybe "can be attributed to".

Page 10, Line 33: change "use" to "uses"

Page 11, Line 15: change "larger in magnitude as going toward" to "increases in magnitude toward"

Page 11, Line 30: change "These differences of CONV" to "The differences in CONV"

Page 12, Line 19: change "should" to "could" (since model biases are potentially possible in CONV as well, given lack of non-satellite data in stratosphere).

---

## Referee Comment (RC2) · Anonymous Referee #2 · 6 Jun 2017

This is an interesting paper discussing the differences between 3 versions of the JRA reanalysis system; a standard version without assimilation of satellite information, a version with satellite information included, and a AMIP type version. The focus is on differences in stratospheric variability and in particular on differences in the number and the characteristics of sudden warmings.

The warmings are diagnosed both with the zonal mean wind at 60 N, 10 hPa and the geopotential height at 10 hPa giving the displacement and shape of the vortex.

The two versions with assimilation give very similar results although inclusion of the satellite data seems to increase the stratospheric variability somewhat. The AMIP type

experiment, on the other hand, shows much less stratospheric variability and very few sudden warmings.

The English could be somewhat improved in places but otherwise the paper is clear and the conclusions well argued for. I will recommend that the paper is accepted after some relatively minor changes.

Major comments:

I think the summary and discussion should be expanded a bit. What does it mean for the quality of the reanalysis if the AMIP type experiment has too little variability? I guess you would expect that if the model was perfect then the statistics of the AMIP experiment would be similar to the statistics of the reanalysis. Do the results of this paper mean that we should expect the number of sudden warmings to be underestimated in the reanalysis products?

It is generally accepted that sudden warmings include some preconditioning. It could be the case that the general lower wave-forcing in the AMIP experiment leaves the vortex very strong so that even after a strong wave event we don't see a sudden warming. Perhaps this is discussed in the text with other words but I would like the authors to include the concept of "preconditioning".

Minor comments:

Perhaps the title could be more precise on what the paper is about. I would suggest that the words "reanalysis" and "assimilation" should be included in the title. Not everybody knows what the "JRA family" is.

p1, l15: This sentence is unclear.

p2, l16: ".. will also be investigated". Does that refer to the S-RIP or the present paper?

p5, top: The degrees of freedom used in the statistical tests should be given. I guess you treat all months as independent. But this assumption at least requires some discussion.

p11, l3: over 4 -> above 4

p17, l8 RMSE -> RMSD?

p5, l22: The sentence "The zonal wind at the gridpoints " does not make sense.

p5 l30: It should be made clear here that this is sentence refers to a single event.

p6, l6: The difference in the widths of the pdf's is very small.

p6, l28: Why is there this linear relationship between AR and CL. It seems that if the displacement is large then the change in shape is large? Is this physically based or an effect of the way the parameters are calculated?

Table 1: Why is the AMIP experiment not in the table?

---

## Referee Comment (RC3) · Anonymous Referee #3 · 8 Jun 2017

**Review: Comparison of large-scale dynamical variability in the extratropical stratosphere among the JRA-55 family data sets**

Taguchi

Atmos. Chem. Phys. Discuss.

June 8, 2017

**Summary**

In this study, the author describes an interesting and novel analysis which compares large-scale stratospheric variability in the three datasets which make up the JRA-55 family. This means he can directly compare (mostly SSW) cases in the full data assimilation system and in the case with no satellite measurements and when there are no atmospheric observations at all. I have not seen anyone attempt this kind of analysis before, and it makes a very useful contribution to our understanding of stratospheric variability and SSWs in particular. That said, and while I think the paper should be published, I also thought that the paper failed to fully capitalise on the potential of the approach. I try to outline my major concerns below. Similarly, in its current form the paper is probably a little too long and could benefit from some careful editing to bring out the main results (and remove some of the current figures).

**Major comments**

Style   The paper takes a largely descriptive approach to comparing the datasets until section 6. I really was hoping that the author might incorporate a little more discussion throughout the manuscript, particularly when comparing the JRA-55 and JRA-55C datasets, since there is a great deal more that might be said about the differences here, particularly since they seem to be confined to the upper stratosphere and sometimes significantly influence the timing of SSW events. As noted above, the paper describes the comparisons in quite a lot of detail leading to unnecessary figures and text and making the paper overly long.

Methodology   In a number of places the author compares the SSWs detected in the standard JRA-55 dataset with those detected in the other two datasets. I can see the merit in this, but wouldn't it make more sense to compare the same dates when SSWs are detected in the standard re-analysis with the other datasets. Then one can really assess the extent to which tropospheric information contribute to producing the SSW seen in the standard assimilation. Similarly, I think it could be very informative to compare the JRA55-C and JRA55-AMIP simulations in this way.

· I didn't find the following figures (and associated analysis) added much and suggest they could be removed or added to a supplement. Figures 6, 7, 9, 11, 13.

**Minor comments**

Throughout  I find the name 'conventional' a little confusing for JRA55-C since this is really just 'non-satellite'

p2 l15  Could you say more about the AMIP-style integration? Is this a continuous run with just SST and other boundary forcing?

p3 l15  Why did the author introduce new nomenclature for the datasets? I think using the standard names would be more useful for comparison with other studies.

p8 l3  What do you mean by 'envelope'

Fig. 2 etc.  Please remove the wedge where the shading and contours don't wrap around the zero line

Fig. 5  I found the lines between the two dates (observed and observed in JRA55-C) tended to obscure the points. I think these could be removed without much loss of information.

Fig. 8  The arrows and labels in the panels made the quite hard to read and to distinguish different points. I like this plot but could it be simplified?

Fig. 10  Could you show the heatflux distribution at the bottom of each panel for the different cases to make the point made in the text about the shift between the two datasets more clearly?

---

## Author Comment (AC1) · 27 Jul 2017

Author response to Referee 1 comments

I thank the referee for reviewing the manuscript. I have tried to address all of the comments and revise the manuscript, and hope the revision is satisfactory. Before responding to each comment, I note several changes:

-Figures 10 and 11 now include results from CONV.

-Figure 12 in the previous version has been removed. The figure and discussion were a bit lengthy compared to the messages obtained. I mention the results from the figure in the text only, so that this does not affect the overall argument in this study.

-I have corrected some existing figures (Figs. 2c, 5b, 9a,b), but this does not affect the argument. Figure 5b in the previous version used the (CL, AR) data for STDD on the MSSW onset dates identified in CONV. It now uses the (CL, AR) data for STDD on the MSSW onset dates identified in STDD. In Figs. 2c and 9a,b, a few color shades did not appear as intended.

Please also note that the pages and numbers in my response below refer to those in the (unformatted) manuscript.

One general but minor comment I have is that the use of the STDD, CONV, and AMIP acronyms was not particularly useful and in fact made things more muddled. I would recommend just sticking with JRA-55, JRA-55C, and JRA-55AMIP. If the author thinks the acronyms add something, then the acronyms should at least be used earlier in the abstract/introduction so that readers become used to seeing them prior to section 2.

Because this study deals with the JRA-55 family data only, the "JRA-55" part in JRA-55 (standard), JRA-55C, and JRA-55AMIP are obvious and redundant, and hence could be removed. One could then refer to the products as standard, conventional, and AMIP: I simplify these by using the acronyms (STDD, CONV, and AMIP), and I think these are clear and easy to remember when reading the manuscript. I have introduced these acronyms in Abstract and Section 1, as suggested.

The other general comment I have is that I appreciated the creativity the author used to make the plots, some of which were very useful ways of visualizing the results; but Figure 8 went a bit overboard, with different symbols, colors, letters and numbers. Some of these attributes may be useful, like the colors; but the different symbols didn't seem to point to anything of particular interest. Right now the text reads (Page 8, Line 21-22 and Line 27-28): "Both overestimations and underestimations in the zonal wind by CONV occur for these cases", which indicates that these symbols are not adding much to our understanding and making the plot more difficult to read. The author could just make a note in the text that they also considered biases in the CONV winds relative to the STDD winds but didn't notice any systematic relationship.

As suggested, I have simplified Fig. 8 and added a note in the text (p. 8, l.23).

Specific Comments

Page 1, Line 9, 19: It is not entirely clear what "conventional" observations include until section 2. It would be better to directly state here that this means everything except satellite data.

Fixed.

Page 1, Line 13: Readers unfamiliar with vortex geometry diagnostics may be unfamiliar with "aspect ratio" of the polar vortex; might be good to briefly relate this to displacements/splits here to be clear. Also in section 2.2.2.

I have added "in which the vortex is highly stretched or splits" to clarify MSSW features of high aspect ratio of the polar vortex (p. 1, l. 15). In Section 2.2.2, we already mentioned that AR is a measure of how stretched the vortex is. A more specific relationship of CL and AR is explained when presenting results (p. 6, l. 30-).

Page 2, Line 17: Here, is "This study" referring to S-RIP, or to the current study?

Fixed. I have rephrased this part as "The present study".

Page 3, Line 21: prior to 1979, how different are the CONV and STDD products?

CONV is different from STDD even before 1979, as Kobayashi et al. (2014) state "The JRA-55C covers the period from November 1972, when the JRA-55 starts to use satellite data, to 2012". Figure 12 does not suggest a gap in RMSD between before and after 1979 for NH winter or SH spring.

Page 5, Line 9-10: Might mention that the tropospheric jets seem shifted, which may (or may not) be consistent with biases in the stratosphere.

I have added a sentence about the zonal wind differences in NH for DJF (p. 5, l. 11-).

Page 6, Lines 32-33, Figure 5: Maybe I'm just confused about what exactly this plot is showing, but I don't understand why in panel (b), the black dots representing STDD don't seem to match the location of the STDD dots in panel (a). This makes lines 32-33 also confusing, as the statement does seem accurate based on the red/black dots in panel (b) but does not seem accurate when comparing panel (b) to panel (a).

Figure 5b is now corrected to use the (CL, AR) data for STDD on the MSSW onset dates identified in STDD. The description about Fig. 5b is modified at p. 7, l. 3.

Page 7, Line 5-6: Might mention that this implies few splits in AMIP.

Fixed. I have added a sentence mentioning this.

Page 8, Line 19-20: the closeness to the y-axis may indicate a large contribution from the wave component, but it terms of proximity to the y-axis, there seems to be many "red"/weak wind points as

well as strong ones. Might need to rephrase to clarify that for large values of the y-component, the zonal wind is stronger.

I have explained that the zonal wind in STDD is relatively strong when the y-component is large ($\gtrsim$ 100 m) at p. 8, l. 21.

Page 9, Line 5: Just to clarify, in Figure 9 are you comparing the products using dates from STDD for both, or the central dates of each product?

We take the former option in Fig. 9. Namely, it uses the dates defined in STDD.

Page 10, Line 12-13: Just to clarify, in Figure 11 are the grey lines all the cases between 25-75th percentile?

I guess "the grey lines" may be thin blue or red lines in Fig. 11. Anyway, yes, this figure used all the $25^{th}$ to $75^{th}$ percentile values of the maximum heat flux in STDD, but the thin lines are removed for simplicity, and only the composites are shown.

Also, is the number of cases in your stated 25-75th percentile range similar for AMIP and STDD? It looks somewhat similar in Figure 10 but might be worth mentioning.

I have added the values for AMIP (p. 10, l. 19).

Also you could consider drawing in light dashed lines the 25th and 75th percentile lines in Figure 10, as a way to clarify what you are plotting in Figure 11.

Fixed.

Page 12, Line 10-12: Is there a difference in latitude of maximum heat flux/EPFD forcing, or a difference in the maximum winds of the climatological polar jet in these data sets (or the edge of the vortex)? Could that also have an influence?

The climatological polar night jet in AMIP is stronger than in STDD, and the maximum is located somewhat more poleward (see Fig. A1, attached below). This feature is also seen in Fig. 1c. On the other hand, the latitudinal profiles of the maximum heat flux are similar between STDD and AMIP. The stronger climatological jet in AMIP may play a role in the different vortex responses (as discussed in Section 5) of AMIP from STDD

Page 15, Table 1 caption: make clear that "differences" are the number of days between onset dates.

Fixed.

Page 16, Figure 1: It wasn't entirely clear from this figure whether values between -5 to +5 are not significant (since they still are shaded)- could this be more clear? Or are any shaded values significant? I guess it's confusing since the colorbar doesn't have any white, non-significant level.

Yes, any shaded values in Fig. 1 are judged to be significant, as "Color shades (in Fig. 1) plot only differences of CONV or AMIP from STDD that are judged to be statistically significant". I have added the word "only" here (p. 4, l. 28).

Page 18-19, Figures 3-4: I think these could be made more clear by staying consistent across panels and always making STDD line dashed, even in panel (a),(d). Also state in the caption what the non-bold and bold lines refer to.
Fixed.

Page 19, Figure 4: I would add in DJF to the plot titles for the top row and SON to the plot titles for the bottom row.
Fixed.

Page 26, Figure 11: by residual [v], do you mean the TEM term v-bar star? Or what is meant by residual [v]? (and page 10, line 17).
Yes, the "residual [v]" means the TEM term v-bar star. I have explained this at p. 10, l. 24, and Fig. 11 caption.

Technical Corrections

Page 1, Line 18: change "It shows" to "We find"
Fixed (p. 1, l. 20).

Page 1, Line 21: "vital" is maybe not quite the right word here
I have rephrased the word to "important" (p. 1, l. 23).

Page 1, Line 26: "Some of weak conditions"! remove "of"
Fixed (p. 1, l. 28).

Page 2, Line 5: "the middle atmosphere science"; remove "the", change to middle atmosphere dynamics
Fixed.

Page 2, Line 10: Missing "with" after "associated"
Fixed.

Page 2, Line 11: "metrological" should be "meteorological"
Fixed.

Page 2, Line 13: I would change "go along with" to "are part of"

Fixed.

Page 2, Line 33: I'm not sure what is meant by "articulate design". Maybe "meticulous" would work better? Same with Page 11, Line 25.

Rephrased as suggested.

Page 5, Line 22: change "at the grid points are" to "at this location is"

Fixed.

Page 5, Line 32: delete "the", change "wind" to "winds"

Fixed.

Page 7, Line 23: should be Fig 6b,d.

Corrected.

Page 8, Line 3: change "whereas even in" to "and even" (remove "in")

Fixed.

Page 8, Line 10: change to "Figure 8 presents scatter plots of the zonal and wave components contributing to the total RMSD values for these cases."

Fixed.

Page 8, Line 12: Change "on" to "of"

Fixed.

Page 9, Line 20-21: change to "explained in terms of wave forcing from the troposphere and/or the vortex response in the stratosphere."

Fixed.

Page 10, Line 13: change to "25th" and "75th"

Fixed.

Page 10, Line 16: might say "This feature is quite similar between the two products by construction" since you have chosen the range of heat flux values to be similar.

Fixed.

Page 10, Line 23: not sure "contributed by" is the right phrase here. Maybe "can be attributed to".
Fixed.

Page 10, Line 33: change "use" to "uses"
This part has been removed (Fig. 12 in the previous version).

Page 11, Line 15: change "larger in magnitude as going toward" to "increases in magnitude toward"
Changed as suggested (p. 11, l. 10).

Page 11, Line 30: change "These differences of CONV" to "The differences in CONV"
Changed as suggested (p. 11, l. 25).

Page 12, Line 19: change "should" to "could" (since model biases are potentially possible in CONV as well, given lack of non-satellite data in stratosphere).
Changed as suggested (p.12, l. 18).

[Figure]

Figure A1: (a) Latitudinal distributions of the climatological zonal mean zonal wind at 10 hPa: (black) STDD, (blue) CONV, and (red) AMIP. Panel (b) is similar, but plots poleward eddy heat flux of waves 1-3 at 100 hPa averaged from lag= -20 to 0 day. Here, the lag= 0 day is when the 21-day mean heat flux in 40-90N, 100 hPa maximizes for each DJF season (as in Fig. 10b).

---

## Author Comment (AC2) · 27 Jul 2017

Author response to Referee 2 comments

I thank the referee for reviewing the manuscript. I have tried to address all of the comments and revise the manuscript, and hope the revision is satisfactory. Before responding to each comment, I note several changes:

-Figures 10 and 11 now include results from CONV.

-Figure 12 in the previous version has been removed. The figure and discussion were a bit lengthy compared to the messages obtained. I mention the results from the figure in the text only, so that this does not affect the overall argument in this study.

-I have corrected some existing figures (Figs. 2c, 5b, 9a,b), but this does not affect the argument. Figure 5b in the previous version used the (CL, AR) data for STDD on the MSSW onset dates identified in CONV. It now uses the (CL, AR) data for STDD on the MSSW onset dates identified in STDD. In Figs. 2c and 9a,b, a few color shades did not appear as intended.

Please also note that the pages and numbers in my response below refer to those in the (unformatted) manuscript.

Major comments:

I think the summary and discussion should be expanded a bit. What does it mean for the quality of the reanalysis if the AMIP type experiment has too little variability? I guess you would expect that if the model was perfect then the statistics of the AMIP experiment would be similar to the statistics of the reanalysis. Do the results of this paper mean that we should expect the number of sudden warmings to be underestimated in the reanalysis products?

I have added a discussion sentence suggesting that even STDD (or CONV) may underestimate stratospheric variability (p. 11, l.27).

It is generally accepted that sudden warmings include some preconditioning. It could be the case that the general lower wave-forcing in the AMIP experiment leaves the vortex very strong so that even after a strong wave event we don't see a sudden warming. Perhaps this is discussed in the text with other words but I would like the authors to include the concept of "preconditioning".

I have added a sentence that suggests a possible relationship of the stronger vortex to a lack of preconditioning in AMIP (p. 12, l. 9).

Minor comments:

Perhaps the title could be more precise on what the paper is about. I would suggest that the words "reanalysis" and "assimilation" should be included in the title. Not everybody knows what the "JRA family" is.

I have added a subtitle "Impacts of assimilation of observational data in JRA-55 reanalysis data" as suggested.

p1, l15: This sentence is unclear.

I have rephrased a few sentences around it (p. 1, l.15-17).

p2, l16: ".. will also be investigated". Does that refer to the S-RIP or the present paper?

I have added a word "there" at the end of sentence, so that this sentence refers to the S-RIP work, not the present study (p.2, l. 17).

p5, top: The degrees of freedom used in the statistical tests should be given. I guess you treat all months as independent. But this assumption at least requires some discussion.

I have mentioned that the degree of freedom in the test is equated to the number of years (p. 5, l.2). This is a reasonable treatment since the test uses DJF or SON means for the target years and they can be regarded independent from year to year.

p11, l3: over 4 -> above 4

This part has been removed.

p17, l8 RMSE -> RMSD?

Fixed (p. 8, l. 18).

p5, l22: The sentence "The zonal wind at the gridpoints" does not make sense.

I have rephrased it (p. 5, l. 24).

p5 l30: It should be made clear here that this is sentence refers to a single event.

I have rephrased the sentence slightly, so that it is clearer that it refers to a single event of the SH MSSW in September, 2002 (p. 6, l.1).

p6, l6: The difference in the widths of the pdf's is very small.

I have rephrased the sentence avoiding the use of "narrow(er)" at p. 6, l. 9-10.

p6, l28: Why is there this linear relationship between AR and CL? It seems that if the displacement is large then the change in shape is large? Is this physically based or an effect of the way the parameters are calculated?

Please note that a large displacement of the vortex is captured by a large decrease in CL. The next sentence also explained/explains how the linear relationship between AR and CL arises for the MSSWs (p. 7, l. 1-2).

Table 1: Why is the AMIP experiment not in the table?

The MSSW dates in AMIP were/are not included in Table 1, since it will not make much sense to compare the AMIP dates to the STDD or CONV dates. The MSSWs in AMIP (free-running simulation) are internally generated, and their exact dates are irrelevant from those in STDD and CONV.

---

## Author Comment (AC3) · 27 Jul 2017

**Author response to Referee 3 comments**

I thank the referee for reviewing the manuscript. I have tried to address all of the comments and revise the manuscript, and hope the revision is satisfactory. Before responding to each comment, I note several changes:

-Figures 10 and 11 now include results from CONV.

-Figure 12 in the previous version has been removed. The figure and discussion were a bit lengthy compared to the messages obtained. I mention the results from the figure in the text only, so that this does not affect the overall argument in this study.

-I have corrected some existing figures (Figs. 2c, 5b, 9a,b), but this does not affect the argument. Figure 5b in the previous version used the (CL, AR) data for STDD on the MSSW onset dates identified in CONV. It now uses the (CL, AR) data for STDD on the MSSW onset dates identified in STDD. In Figs. 2c and 9a,b, a few color shades did not appear as intended.

Please also note that the pages and numbers in my response below refer to those in the (unformatted) manuscript.

**Major comments**

Style: The paper takes a largely descriptive approach to comparing the datasets until section 6. I really was hoping that the author might incorporate a little more discussion throughout the manuscript, particularly when comparing the JRA-55 and JRA-55C datasets, since there is a great deal more that might be said about the differences here, particularly since they seem to be confined to the upper stratosphere and sometimes significantly influence the timing of SSW events. As noted above, the paper describes the comparisons in quite a lot of detail leading to unnecessary figures and text and making the paper overly long.

The descriptive style of this study is by intention, since we compare dynamical variability in the extratropical stratosphere especially from a morphological point of view. I think the present analysis and manuscript are reasonably organized: survey climatology and variability (MSSWs) among the three products in Section 3, describe RMSD distributions and compare them to the MSSWs in Section 4 (STDD vs CONV), and examine wave forcing and vortex response in Section 5 (STDD vs AMIP).

It will be possible to compare vertical structures of the polar vortex between STDD and CONV for various dynamical conditions. Then, one may find that the vortex has largely different structures especially in the upper stratosphere between STDD and CONV even when the two products have a MSSW on a same onset date. However, it will be difficult to combine such an analysis to the current manuscript, and I therefore retain the overall structure (and most figures, except for Fig. 12) of the manuscript. I have added a discussion paragraph to mention this extension as possible future work (p. 12, 1. 20-24).

Methodology: In a number of places the author compares the SSWs detected in the standard JRA-55

dataset with those detected in the other two datasets. I can see the merit in this, but wouldn't it make more sense to compare the same dates when SSWs are detected in the standard re-analysis with the other datasets. Then one can really assess the extent to which tropospheric information contribute to producing the SSW seen in the standard assimilation. Similarly, I think it could be very informative to compare the JRA55-C and JRA55-AMIP simulations in this way.

While Table 1 compares the MSSW onset dates identified in STDD to those identified in CONV and I mention their time differences in places in the text, I also compared/compare features between STDD and CONV on the MSSW onset dates identified in STDD, such as in Figs. 8 and 9f-j. Figures 10 and 11 also use the maximum of the 21 day mean heat flux for DJF identified in each product.

Figure 11, which now includes results from CONV, will contribute to the latter point. It shows that AMIP is an outlier in terms of the zonal wind evolution, while all three products are similar in the other quantifies: AMIP underestimates a vortex response, or a zonal wind deceleration at 10 hPa to a wave forcing of similar strength from the troposphere (which can be regarded as a model bias). STDD and CONV are similar in all quantities of interest, but the zonal wind in CONV is slightly stronger than that in STDD. This seems consistent with the result that CONV misses or delays several MSSWs compared to STDD.

I didn't find the following figures (and associated analysis) added much and suggest they could be removed or added to a supplement. Figures 6, 7, 9, 11, 13. Please refer to my response to the first comment.

**Minor comments**

Throughout I find the name 'conventional' a little confusing for JRA55-C since this is really just 'non-satellite'

I follow the paper (Kobayashi et al. 2014) of the developers of the JRA-55C product for the use of the word "conventional". I also explain the meaning of "conventional" at the beginning (p. 1, l. 10).

p2 115 Could you say more about the AMIP-style integration? Is this a continuous run with just SST and other boundary forcing?

I have added an explanation about AMIP at p. 3, l. 19-20.

p3 115 Why did the author introduce new nomenclature for the datasets? I think using the standard names would be more useful for comparison with other studies.

Because this study deals with the JRA-55 family data only, the "JRA-55" part in JRA-55 (standard), JRA-55C, and JRA-55AMIP are obvious and redundant, and hence could be removed. One could then refer to the products as standard, conventional, and AMIP: I simplify these by using the acronyms (STDD, CONV, and AMIP), and I think these are clear and easy to remember when reading the manuscript.

p8 13 What do you mean by 'envelope'

I have removed the phrase using the word "envelope", as the sentence is clear without it.

Fig. 2 etc. Please remove the wedge where the shading and contours don't wrap around the zero line I have improved the figures as suggested.

Fig. 5 I found the lines between the two dates (observed and observed in JRA55-C) tended to obscure the points. I think these could be removed without much loss of information.

I think that the lines are useful since it enables one to compare/connect corresponding cases between STDD and CONV. I have changed the lines and STDD data points into grey, so that they do not disturb the main distribution for CONV in (b).

Fig. 8 The arrows and labels in the panels made the quite hard to read and to distinguish different points. I like this plot but could it be simplified? I have somewhat simplified the figure.

Fig. 10 Could you show the heat flux distribution at the bottom of each panel for the different cases to make the point made in the text about the shift between the two datasets more clearly? I have changed the figure accordingly.